# Identification of Novel Mutations and Expressions of *EPAS1* in Phaeochromocytomas and Paragangliomas

**DOI:** 10.3390/genes11111254

**Published:** 2020-10-24

**Authors:** Farhadul Islam, Suja Pillai, Vinod Gopalan, Alfred King-Yin Lam

**Affiliations:** 1Institute for Glycomics, Griffith University, Gold Coast, QLD 4222, Australia; farhad_bio83@ru.ac.bd; 2Department of Biochemistry and Molecular Biology, University of Rajshahi, Rajshahi 6205, Bangladesh; 3Faculty of Medicine, School of Biomedical Sciences, University of Queensland, Brisbane, QLD 4072, Australia; s.pillai@uq.edu.au; 4Cancer Molecular Pathology, School of Medicine, Gold Coast, QLD 4222, Australia; v.gopalan@griffith.edu.au

**Keywords:** endothelial PAS domain-containing protein 1 (*EPAS1*), phaeochromocytoma, paraganglioma, genetics, mutations

## Abstract

Endothelial PAS domain-containing protein 1 (*EPAS1*) is an oxygen-sensitive component of the hypoxia-inducible factors (HIFs) having reported implications in many cancers by inducing a pseudo-hypoxic microenvironment. However, the molecular dysregulation and clinical significance of *EPAS1* has never been investigated in depth in phaeochromocytomas/paragangliomas. This study aims to identify *EPAS1* mutations and alterations in DNA copy number, mRNA and protein expression in patients with phaeochromocytomas/paragangliomas. The association of molecular dysregulations of *EPAS1* with clinicopathological factors in phaeochromocytomas and paragangliomas were also analysed. High-resolution melt-curve analysis followed by Sanger sequencing was used to detect mutations in *EPAS1*. *EPAS1* DNA number changes and mRNA expressions were examined by polymerase chain reaction (PCR). Immunofluorescence assay was used to study *EPAS1* protein expression. In phaeochromocytomas, 12% (*n* = 7/57) of patients had mutations in the *EPAS1* sequence, which includes two novel mutations (c.1091A>T; p.Lys364Met and c.1129A>T; p.Ser377Cys). Contrastingly, in paragangliomas, 7% (*n* = 1/14) of patients had *EPAS1* mutations and only the c.1091A>T; p.Lys364Met mutation was detected. In silico analysis revealed that the p.Lys364Met mutation has pathological potential based on the functionality of the protein, whereas the p.Ser377Cys mutation was predicted to be neutral or tolerated. The majority of the patients had *EPAS1* DNA amplification (79%; *n* = 56/71) and 53% (*n* = 24/45) patients shown mRNA overexpression. Most of the patients with *EPAS1* mutations exhibited aberrant DNA changes, mRNA and protein overexpression. In addition, these alterations of *EPAS1* were associated with tumour weight and location. Thus, the molecular dysregulation of *EPAS1* could play crucial roles in the pathogenesis of phaeochromocytomas and paragangliomas.

## 1. Introduction

Phaeochromocytomas and paragangliomas are rare catecholamines producing neural crest tumours derived from neuroendocrine chromaffin cells [1]. Phaeochromocytoma arises in the adrenal gland and paraganglioma in extra-adrenal chromaffin cells outside the adrenal gland [2,3]. The most common extra-adrenal location of this tumour is in the carotid body [3,4]. As the behaviour of this group of tumours is difficult to predict, the World Health Organization now classified the tumours as metastasizing and non-metastasizing (instead of benign and malignant) [5].

The genetics of phaeochromocytomas and paragangliomas is complex. Approximately 40% of phaeochromocytomas/paragangliomas are familial and part of different hereditary syndromes such as multiple endocrine neoplasia type 2 (MEN2), von Hippel-Lindau disease (VHL), neurofibromatosis type 1 (NF1) or familial phaeochromocytoma-paraganglioma syndrome [1,6,7]. The others are apparently sporadic [8,9]. Genes associated with the pathogenesis of phaeochromocytomas/paragangliomas are divided into two major clusters depending on their gene expression profile [6,8]. Cluster 1 genes are involved with the pseudo hypoxic pathway of tumour development, e.g., *HIF2A*, *PHD2*, *VHL*, *SDHx*, *IDH*, *MDH2*, *SLC25A11*, *DLST* and FH [10]. Cluster 2 involves genetic mutations associated with the abnormal activation of kinase signalling pathway, e.g., *NF1*, *KIF1Bβ*, *MAX*, *RET*, *TMEM127* [9]. Additionally, activating mutations of Wnt-signalling pathway or its components, including somatic mutations of *CSDE1* (cold shock domain containing E1) and *MAML3* (mastermind like transcriptional coactivator 3) genes are known as cluster 3 in phaeochromocytomas and paragangliomas [11]. Overall, the genetic heterogeneity of phaeochromocytomas and paragangliomas is very puzzling, and this makes the genetic diagnosis, counselling and clinical follow-up of patients with these tumours more challenging.

Recent studies have demonstrated that mutations in endothelial PAS domain-containing protein 1 (*EPAS1*) is associated with manifestations of phaeochromocytomas and paragangliomas [1,9,12,13,14]. *EPAS1*, also known as *HIF-2α*, encodes for one of the hypoxia inducible factor (HIF) family members, which is involved in the hypoxic response [15]. The identification of somatic mutations affecting *EPAS1* in phaeochromocytomas and paragangliomas led to the hypothesis that the stabilization of HIF-α plays crucial roles in the development of chromaffin tumours [12,14] through the phenomenon known as pseudo-hypoxia. In a pseudo-hypoxic state, abnormal HIF-α function enhances cell proliferation and tumour growth [16,17]. So far, *EPAS1* mutations have been reported in a few phaeochromocytomas and paragangliomas [1,12]. Therefore, further research on the effects of *EPAS1* mutation on gene expression in a large cohort of patients with phaeochromocytomas and paragangliomas is critical to unveil its roles in the phaeochromocytomas/paragangliomas pathogenesis. Herein, we investigated mutations in *EPAS1* in a cohort of phaeochromocytoma (*n* = 57) or paraganglioma (*n* = 14) and have described the clinical, molecular and genetic features of patients carrying somatic *EPAS1* mutations. In addition, *EPAS1* DNA number variation, mRNA, protein expressions and their correlation with clinical and pathological features of patients with phaeochromocytomas and paragangliomas were also investigated.

## 2. Materials and Methods

### 2.1. Recruitments of Tissues and Sample Selection

Tumour tissues from patients who underwent resection of phaeochromocytomas (*n* = 57) and paragangliomas (*n* = 14) were collected from hospitals in Australia and Hong Kong during the period of 1973 to 2015. These patients were recruited prospectively with no selection bias. They were excluded from the study in cases of lacking adequate tumour tissue sampled or missing clinical data. Ethical approval for this work has been obtained from the Griffith University Human Research Ethics Committee (GU Ref No: MED/19/08/HREC and MSC/17/10/HREC).

The resected phaeochromocytomas/paragangliomas tissues were fixed in 10% formalin and embedded in paraffin wax for downstream analysis. Histological sections from these paraffin blocks were cut using microtome (Leika Biosystem, Wetzlar, Germany). The sections were then stained with haematoxylin and eosin (H&E) and examined under a light microscope for identifying the histopathological features of patients with phaeochromocytomas/paragangliomas by the author (A.K.L.). After reviewing the histological sections, a block was chosen from each of the phaeochromocytomas and paragangliomas from 71 patients (34 women; 37 men) and 10 non-neoplastic adrenal tissues, which act as controls (adrenal glands with no cancer collected from patients resected for renal cell carcinomas during the operation as a part of the procedure), to be included in this study. The selection of block from each case was based on having adequate tumour tissue portion (>70% area occupied by the tumour). The adrenal medulla tissue from non-neoplastic adrenal glands were micro dissected out for RNA and DNA extractions.

### 2.2. Clinical Data of the Selected Cases

The study population includes 56 Chinese patients from Hong Kong and 15 patients from Australia of European descent. The mean age of the patients with phaeochromocytomas and paragangliomas used in this study was 45 years (range, 2 to 79). There were 57 patients with phaeochromocytomas and 14 with paragangliomas. All the 14 patients with paraganglioma (located at the carotid body) and 8 patients with phaeochromocytomas do not have catecholamines detected clinically. Moreover, six patients with phaeochromocytomas harbouring *EPAS1* mutation had shown catecholamine secretion. In addition, 4 had multiple endocrine neoplasia 2 (MEN 2) and 2 had neurofibromatosis. Of the 4 patients with MEN 2, one had bilateral phaeochromocytomas and one had two tumours in the same adrenal gland. Apart from these, there are 4 patients with bilateral phaeochromocytomas but with no known genetic predispositions. In addition, three patients have bilateral carotid paragangliomas. Furthermore, one patient had breast carcinoma and one had myocardial infarction.

The mean follow-up and median follow-up of the patients with phaeochromocytomas/paragangliomas were 70.5 months and 54 months, respectively. Of these patients, 8 has metastasizing diseases. The interval between the date of surgery for tumours and the date of death or the closing date of the study was defined as the follow-up period in this study. The patients’ actuarial survival rate was calculated from the date of surgical resection of phaeochromocytomas/paragangliomas to the date of death or last follow-up. The endpoint in the statistical analysis is defied by tumour-related death. A schematic flow chart of the experiments is shown in Figure 1.

### 2.3. Extractions of DNA and RNA

The selected tissues were sectioned using microtome (Leica Biosystem, Wetzlar, Germany) into 7 μm slices for RNA and DNA extractions [18]. The Qiagen DNeasy Blood and Tissue kit (Qiagen Pty. Ltd., Hilden, Germany) was used to extract the DNA according to the manufacturer’s guidelines. The miRNeasy Mini kit (Qiagen Pty. Ltd., Hilden, Germany) was used to extract RNA from the tissue sections according to the manufacturer’s protocol. Measurement of optical density (OD) using a nanodrop spectrophotometer was used to check the purity of the extracted DNA and RNA. The concentrations of DNA and RNA were noted in ng/μL and used for further analysis.

### 2.4. High-Resolution Melt (HRM) Curve Analysis

HRM analysis of exon 1 and exon 9 of *EPAS1* gene sequence were carried out to screen the possible mutations in genomic DNA extracted from 71 tumours and 10 control tissues. The Rotor-Gene Q detection system (Qiagen Pty. Ltd., Hilden, Germany) was used to perform HRM by amplifying target sequences. The Rotor-Gene ScreenClust HRM Software was used to analyse the HRM curve. Exons 1 and 9 of the *EPAS1* sequence were amplified using standard PCR system (10 μL). The reaction mixture consisted of 5 μL 2× sensimix HRM master mix, 1 μL of genomic DNA (30 ng/μL), 2 μL RNase free water and each primer 1 μL. Thermal cycling profile of the PCR was previously described [19]. All PCR experiment were carried out in triplicate including a no template (negative) control. In all assays, by increasing the temperature from 65 °C to 85 °C with a temperature increase rate of 0.05 °C/s and recording fluorescence, the melt curve data were generated.

### 2.5. Confirmation of Mutations by Sanger Sequencing

Sanger sequencing was used to confirm the possible mutations detected by HRM analysis. In short, the susceptible mutated PCR products were purified using the NucleoSpin^®^ Gel and PCR Clean-up kit (Macherey- Nagel, Bethlehem, PA, USA) according to the manufacturer’s protocols followed by HRM analysis. Big Dye Terminator (BDT) chemistry Version 3.1 (Applied Biosystems, Foster City, CA, USA) under standardised cycling PCR conditions was used for sequencing the purified PCR products. A 3730xl Capillary sequencer (Applied Biosystems) was used analyse the generated data at the Australian Genome Research Facility (AGRF). Finally, the sequences were then examined using Sequence Scanner 2 software (Applied Biosystems).

### 2.6. In Silico Analysis

For computational analysis, the Ensembl transcript ID ENST00000263734 (*EPAS1*) was used as input. The identified mutations were computationally analysed using free tools, i.e., Mutation Taster with NCBI 37 and Ensembl 69 database release [20], protein variation effect analyser (PROVEAN) and sorting intolerant from tolerant (SIFT) to predict the impacts of the detected mutations on the protein functionality. In addition, the predicted outcomes were compared with ExAc and 1000 Genomes mutation databases. The cut-off values, −2.5 for PROVEAN and 0.05 for SIFT were used to predict the pathogenic/non-pathogenic mutations in the present study.

### 2.7. Quantitative Real-Time PCR (qPCR) Analysis

DNA copy number changes in phaeochromocytomas/paragangliomas (*n* = 71) and non-neoplastic adrenal (*n* = 10) tissues were examined using QuantStudio 6 Flex Real-Time PCR System (Thermo Fisher Scientific, Waltham, MA, USA). A detailed protocol for this method was previously reported [21,22]. In short, a 20 μL qPCR reactions were performed. The reaction mixture consisted of 10 µL of DyNAmo Flash SYBR green master mix (Bio-Rad), 1.5 µL of each primer (5 μmol/L), 3 μL of DNA (50 ng/μL), and 4 µL of RNase free water.

In addition, first-strand cDNA was generated from total RNA using DyNAmo™ cDNA Synthesis Kits (Qiagen) as previously described [23]. Then, *EPAS1* mRNA expression changes were analysed in phaeochromocytomas/paragangliomas (*n* = 45) and non-neoplastic adrenal tissues (*n* = 10) were studied following a previously published protocol [19,22].

The PCR amplification efficiencies were normalised using multiple housekeeping genes, including beta-actin, 18s and GAPDH (glyceraldehyde 3-phosphate dehydrogenase). Finally, GAPDH was selected based on consistent results. DNA fold changes and mRNA expression were calculated according to the previously published protocol [19,24]. In this study, a fold change of more than 2 was considered as DNA amplification or high *EPAS1* expression, whereas fold change of less 0.5 was considered as DNA deletion or low *EPAS1* expression.

### 2.8. Immunofluorescence

The possible impact of *EPAS1* mutation on protein expression in mutated (*n* = 8) and non-mutated (*n* = 8) tissues samples of phaeochromocytomas and paragangliomas was investigated by immunofluorescence analysis. For this, the tissue sections (4 μm) were de-waxed (xylene) and re-hydrated (alcohols and water). The tissues sections were then blocked with peroxidase block solution (Dako Australia Pty. Ltd., Sydney, NSW, Australia) and after that the sections were incubated with mouse *EPAS1* (1:150) monoclonal antibody (Santa Cruz, Dallas, TX, USA) overnight at 4 °C. Then, the tissue sections were incubated with secondary antibody labelled with fluorescein isothiocyanate (FITC) (Sigma-Aldrich) at room temperature for two hours. Subsequently, the tissue sections were mounted on glass slides. Finally, the slides were examined under a confocal microscope (Nikon A1R+, Nikon Inc., Tokyo, Japan). The generated signals from EPAS1 was categorised as “0” (0% to less than 10%), “+” (10% to <30%), “++” (30% to <50%) and “+++” (>50%) according to the percentage and intensity of EPAS1 protein staining. Tissues in the categories of “0” and “+” were classified as No change or low expression, whereas “++” and “+++” were considered as high *EPAS1* expression.

### 2.9. Statistical Analysis

For statistical analysis, all the data were entered into a computer database and analysed using the Statistical Package for Social Sciences for Windows (version 25.0, IBM SPSS Inc., New York, NY, USA). Variable groups were compared and analysed using the chi-square test, student t-test and Fisher’s exact test. The Kaplan–Meier method used analysed the survival rates of patients. For all the analysis, the significance level was taken at *p* < 0.05.

## 3. Results

### 3.1. Identification of Novel EPAS1 Mutations in Phaeochromocytomas/Paragangliomas

In the present study, genetic alterations in the *EPAS1* sequence were noted in 12% (7/57) of phaeochromocytoma tumours (Table 1). Two novel mutations c.1091A>T and c.1129A>T were identified in exon 9 (Figure 2). Both mutations were somatic heterozygous missense mutations (p.Lys364Met and P.Ser377Cys). In paragangliomas, 7% (1/14) of patients had shown *EPAS1* mutations. The only c.1091A>T (p.Lys364Met) was identified tumour tissues in paraganglioma (Table 1). However, no mutant mutation was detected in non-neoplastic adrenal tissues. In silico analysis predicted that the identified mutation c.1091A>T (p.Lys364Met) was disease causing as it could cause changes in protein structure and function with pathological consequences (Table 1). Contrastingly, the c.1129A>T (P.Ser377Cys) mutation was predicted to be a polymorphism/neutral and was identified in two cases. These identified mutations were not found in the ExAc, PubMed or 1000 Genomes mutation databases.

Patients with phaeochromocytomas bearing mutation were four males and four females with a mean age of 41 years (age range 22–58). The mutations noted in this study were from functioning tumours. Among the seven patients with phaeochromocytoma harbouring mutated *EPAS1*, two patients had clinical confirmation of neurofibromatosis 1 (case 94 and case 122). Table 2 revealed the associations of identified mutations with the clinical and pathological factors of patients with phaeochromocytomas/paragangliomas (Table 2). *EPAS1* mutations occur in tumours with higher tumour weight (>50 gm) (Table 2; *p* = 0.0001). Moreover, *EPAS1* mutations were associated with the large tumour size (≥50 mm). Most of the mutated samples (7 out of 8) had larger tumours (Table 2; *p* = 0.044). Other than this, there was no association between the mutations in *EPAS1* with patients’ age, sex, race or the side, location and presence of tumour metastasis in patients with phaeochromocytomas/paragangliomas.

### 3.2. EPAS1 DNA Number Variations in Phaeochromocytomas/Paragangliomas

Amongst the phaeochromocytomas and paragangliomas (*n* = 71), 79% (56/71) showed DNA number amplification, whereas 21% (15/71) had DNA number deletion. In addition, *EPAS1* DNA number changes (ratio of expression) in phaeochromocytomas/paragangliomas were significantly (*p* < 0.01) higher when compared to that of non-neoplastic adrenal tissues (1.486 ± 0.011 versus 1.186 ± 0.015) (Figure 3A). *EPAS1* DNA number changes correlated with the location of tumour in patients (Table 3). Many patients with tumours located in the adrenal gland (phaeochromocytomas) showed *EPAS1* DNA amplification when compared to those with tumours outside adrenal gland (paragangliomas) (84.21% versus 57.14%; *p* = 0.037). Similarly, phaeochromocytoma showed a significantly lower proportion of *EPAS1* DNA deletion when compared to paragangliomas (15.97% versus 42.86%; Table 3).

### 3.3. EPAS1 mRNA Expressions in Phaeochromocytomas/Paragangliomas

mRNA expressions of *EPAS1* in tumour and non-neoplastic control tissues samples were shown in Figure 3B. The expressions were statistically significant differences between tumour and non-neoplastic control tissues samples (1.64 ± 0.07 versus 1.26 ± 0.03; *p* = 0.002). Among the tumours (phaeochromocytomas, *n* = 37; paragangliomas, *n* = 8), 53% (*n* = 24/45) showed high levels of *EPAS1* mRNA expression, whereas 47% (*n* = 21/45) had *EPAS1* low-level expression. The associations of *EPAS1* mRNA expression with clinicopathological parameters are presented in Table 4. Changes of *EPAS1* mRNA expressions were correlated with tumour location and the metastasizing potential of the tumours. In brief, phaeochromocytomas had higher *EPAS1* mRNA expressions when compared with paragangliomas (Table 4). Moreover, all paragangliomas showed low *EPAS1* mRNA expression, whereas 35% of phaeochromocytomas had low *EPAS1* mRNA expression (Table 4; *p* = 0.001). In addition, 61% (*n* = 22/36) of non-metastasizing tumours had high *EPAS1* mRNA overexpression, whereas 78% (*n* = 7/9) of metastasizing malignant tumours had low *EPAS1* mRNA expression (Table 4; *p* = 0.057).

### 3.4. EPAS1 Protein Expression in Phaeochromocytomas/Paragangliomas

The immunofluorescence staining of *EPAS1* in the representative phaeochromocytomas/paragangliomas and non-neoplastic adrenal tissue samples showed a different degree of staining under confocal microscopy (Figure 4). Among the *EPAS1* mutation-positive tumour samples (phaeochromocytomas, *n* = 7; paragangliomas, *n* = 1), 37.5% showed no change or low expression and 62.5% of samples had high EPAS1 protein expression (Figure 1). On the contrary, in *EPAS1* mutation-negative tumour tissues (*n* = 8), 75% of samples showed no change or low expression and 25% of patients had shown high EPAS1 protein expression (Figure 1).

The median overall follow-up of patients with phaeochromocytomas/paragangliomas was 54 months. There is no significant difference in survival rates among *EPAS1*-mutated, DNA number-changed, and mRNA-altered groups were analysed (*p* > 0.05).

### 3.5. Association of EPAS1 DNA Number Variation, mRNA Expression, Protein Expression and Mutations in Phaeochromocytomas

A statistically significant positive correlation of *EPAS1* DNA copy number changes and mRNA expression were noted in the present study (*r* = +0.538; *p* = 0.009, Fisher exact test). As shown in Figure 5A, 70.2% (40/57) copy number amplified tumour samples had higher *EPAS1* mRNA expression, whereas *EPAS1* mRNA downregulation was only noted in 71.5% (5/7) of the *EPAS1* DNA number deletion tumours (Figure 5A). The associations of *EPAS1* mRNA expressions with copy number variation are presented in Figure 5B. Patients with *EPAS1* DNA copy number amplification exhibited significantly higher mRNA expression in comparison to no change or deletion groups (Figure 5B). Most of the tumour samples with no changes or deletion of *EPAS1* DNA copy number exhibited similar results in mRNA expressions. On the other hand, most of the tumour samples with *EPAS1* copy number amplifications had increased mRNA and protein expression.

DNA copy number variations and mRNA expressions among *EPAS1*-mutated and non-mutated cases were also analysed (Figure 6). *EPAS1*-mutated cases showed significant copy number amplification (Figure 6A) and higher level of mRNA expression (Figure 6B) when compared to those of non-mutated cases (*p* < 0.05). In addition, the majority (5/8) of tumours with *EPAS1* mutations showed higher expression of protein when compared with non-neoplastic adrenal gland or mutation-negative tissues (Figure 4).

## 4. Discussion

This study identified novel mutations and clinicopathological implications of *EPAS1* dysregulation in the pathogenesis of phaeochromocytoma/paraganglioma. The mutation analysis of *EPAS1* in 71 phaeochromocytomas and paragangliomas resulted in the identification of two heterozygous somatic mutations, which have not been previously reported in phaeochromocytomas and paragangliomas. The detected mutation p.Lys364Met in exon 9 was predicted to be pathogenic to the functionality of *EPAS1* protein in computational analysis. Contrastingly, the other mutation p.Ser377Cys was identified as a polymorphism and could act as tolerated or neutral on the protein functionality (Table 1). Moreover, in the present study, we have noted that the patients with phaeochromocytoma/paraganglioma having *EPAS1* mutations had no mutations in phaeochromocytoma/paraganglioma-susceptible gene panels except NF1 in two patients with Neurofibromatosis 1. In addition, most of the mutated samples had shown gain-of-function of *EPAS1*, i.e., *EPAS1* DNA number amplification, high mRNA and protein expression in the present study. Similar results were demonstrated in previous studies [1,12]. Furthermore, gain-of-function mutations (p.Phe374Tyr and p.Met368Ile) in exon 9 *EPAS1* in phaeochromocytomas/paragangliomas associated with the increased stability of HIF2α [12,25]. Higher EPAS1 protein expression in mutated samples implied that the results of the current study are consistent with the previous findings. However, further studies are essential to determine the functional pathogenicity of the identified mutations.

*EPAS1* mutations have been identified in several pathological conditions in humans, including congenital heart disease [26], erythrocytosis [27], Lynch syndrome [28], polycythaemia [29] and in various tumours, e.g., in paraganglioma [30,31], phaeochromocytoma [12], pancreatic adenocarcinoma [32]. Somatic *EPAS1* mutations in different cancers indicate that these mutations may occur in a cell during embryogenesis or later, which in turn predispose the affected tissues or organ to form tumours [14]. In addition to the somatic mutation of *EPAS1*, inherited and constitutional mutations were associated with the pathogenesis of phaeochromocytoma and paragangliomas [1,25]. Moreover, the type of mutations along with additional accumulative genetic, epigenetic and environmental factors are attributed to the pathogenesis of phaeochromocytoma and paraganglioma.

The current study reports *EPAS1* mutations in patients with phaeochromocytomas and paragangliomas and their correlation with various clinicopathological factors. The association of *EPAS1* mutations with high tumour weight (*p* = 0.001) and larger tumour size (*p* = 0.044) implied that mutations of *EPAS1* may contribute to the progression of this group of tumours. The underlying mechanism of *EPAS1*-induced carcinogenesis is poorly understood; however, it is reported that *EPAS1* promote angiogenesis by interacting with both vascular endothelial growth factor (VEGF) and its receptor Fms Related Tyrosine Kinase 1 (Flt1) [33]. Thus, the gain-of-function mutations of *EPAS1* can lead to increased expression of VEGF and Flt1 in endothelial cells, which in turn promotes angiogenesis, thereby promoting tumour growth and progression [33]. It was noted that the suppression of *EPAS1* via shRNA in breast carcinomas cells reduced the cellular response and inhibited angiogenesis significantly, resulting in reduced tumour growth and development [34]. In addition, mutations in *EPAS1* increased the stability of HIF2α leading to pseudo-hypoxic response, thus allowing for the activation of target genes and hence contributing to chromaffin cells tumorigenesis [35]. Therefore, mutations of *EPAS1* may reduce HIF2α breakdown, which in turn could promote carcinogenesis by inducing pseudo-hypoxic conditions and promoting angiogenesis.

DNA copy number aberrations, dysregulated mRNA and protein level expressions in genes are commonly acquired changes in the cancer cells, thus playing a key role in the initiation and progression of cancers [36,37]. In the current study, the aberrant *EPAS1* DNA number in patients with phaeochromocytomas/paragangliomas implied its potential roles in carcinogenesis. Similarly, the dysregulated expression of *EPAS1* mRNA in tumour samples indicated the tumour-associated functionality of *EPAS1* in phaeochromocytomas and paragangliomas. Importantly, the association of *EPAS1* DNA number amplification (*p* = 0.037) and *EPAS1* mRNA expression (*p* = 0.001) with tumour location (adrenal gland versus carotid body) suggested the clinical significance of *EPAS1* in the carcinogenesis of phaeochromocytomas and paragangliomas. The differential *EPAS1* DNA number and mRNA expression in phaeochromocytomas and paragangliomas implied that the aberration of *EPAS1* could affect these tumours in a different manner. The difference in the molecular makeup of adrenal gland and carotid body may have contributed to this statistical significance of *EPAS1* mRNA expression [38]. However, many patients with the metastatic tumours had shown lower *EPAS1* mRNA expression when compared to that of non-metastatic patients (Table 4; *p* = 0.057). Although the statistical difference is above the cut-off value (*p* < 0.05), low metastatic sample size (*n* = 9) could be the contributing factor of this relationship.

The statistical relationship of *EPAS1* DNA amplification and increased mRNA expression in patients with phaeochromocytomas/paragangliomas in this study indicated that hypoxic tumour niche induces molecular alterations of *EPAS1*, which, in turn, can promote carcinogenesis. Moreover, the DNA amplification and mRNA overexpression in patients with phaeochromocytomas/paragangliomas bearing *EPAS1* mutations is indicative of the concerted aberration of *EPAS1* in the pathogenesis for this group of tumours. In addition, higher expressions of *EPAS1* protein in mutated samples indicate tumour-supporting roles of *EPAS1* in the pathogenesis of phaeochromocytoma/paraganglioma. Previous studies reported that dysregulation and gain-of-function mutation of *EPAS1* associated with neuroendocrine tumours such as paraganglioma and phaeochromocytomas by inducing pseudo-hypoxic tumour microenvironment [9,13,39]. Thus, the findings of the current study agree with previous studies.

## 5. Conclusions

This study has reported multiple novel *EPAS1* mutations in patients with phaeochromocytomas/paragangliomas. These mutations were noted to be related with altered expression and/or structural and functional changes in *EPAS1*, which in turn could play an important role in the pathogenesis of phaeochromocytomas and paragangliomas. In addition, the association of *EPAS1* DNA number changes and mRNA expression with clinical and pathological factors, including tumour weight, size, location and the type of tumour, denotes the potential clinical significance of *EPAS1* in predicting disease progression.

## Figures and Tables

**Figure 1 genes-11-01254-f001:**
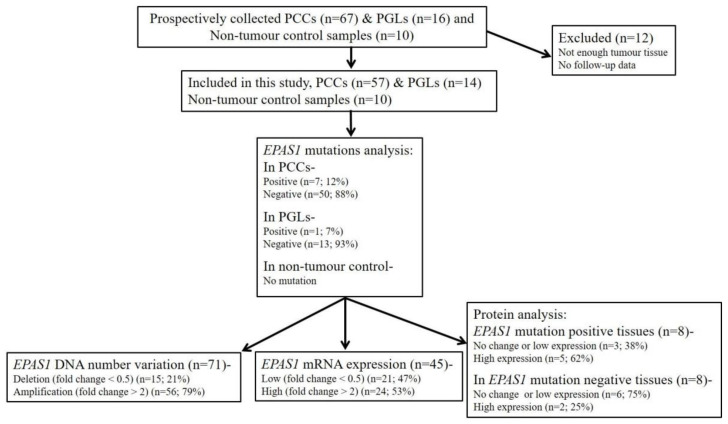
Schematic illustration of the methodological flow used for clinical samples analysis in the present study. PCCs: Phaeochromocytomas; PGLs: paragangliomas.

**Figure 2 genes-11-01254-f002:**
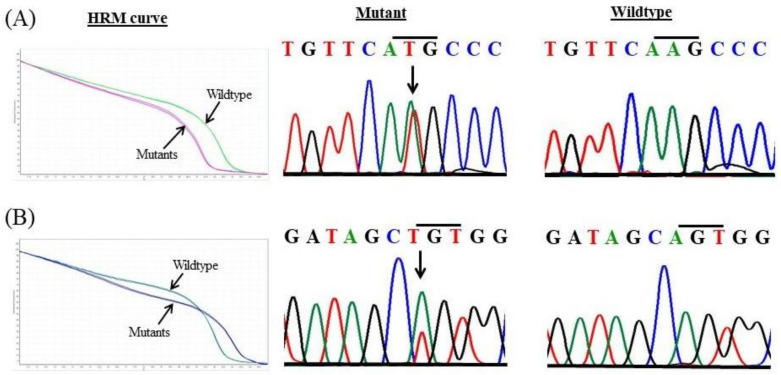
Novel mutations in *EPAS1* identifies in phaeochromocytomas and paragangliomas. Comparison of high-resolution melt (HRM) curve analysis and Sanger sequencing of the mutations identified in patients with phaeochromocytomas/paragangliomas. (**A**) Representative HRM curve and chromatograph for the missense mutation c.1091A>T (p.Lys364Met). (**B**) Representative HRM curve and chromatograph for the substitutional mutation c.1129A>T (p.Ser377Cys).

**Figure 3 genes-11-01254-f003:**
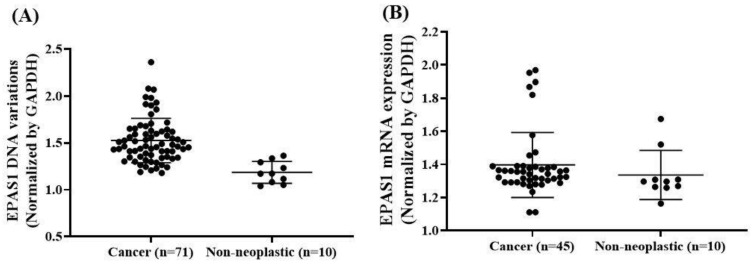
*EPAS1* DNA number and mRNA expression profile in patients with phaeochromocytomas/paragangliomas. (**A**) Patients with phaeochromocytoma/paraganglioma exhibited significant *EPAS1* DNA amplification in comparison to that of non-neoplastic tissues (*p* < 0.01). (**B**) Similarly, patients with phaeochromocytomas/paragangliomas exhibited significant overexpression of *EPAS1* mRNA in comparison to that of non-neoplastic tissues (*p* = 0.002).

**Figure 4 genes-11-01254-f004:**
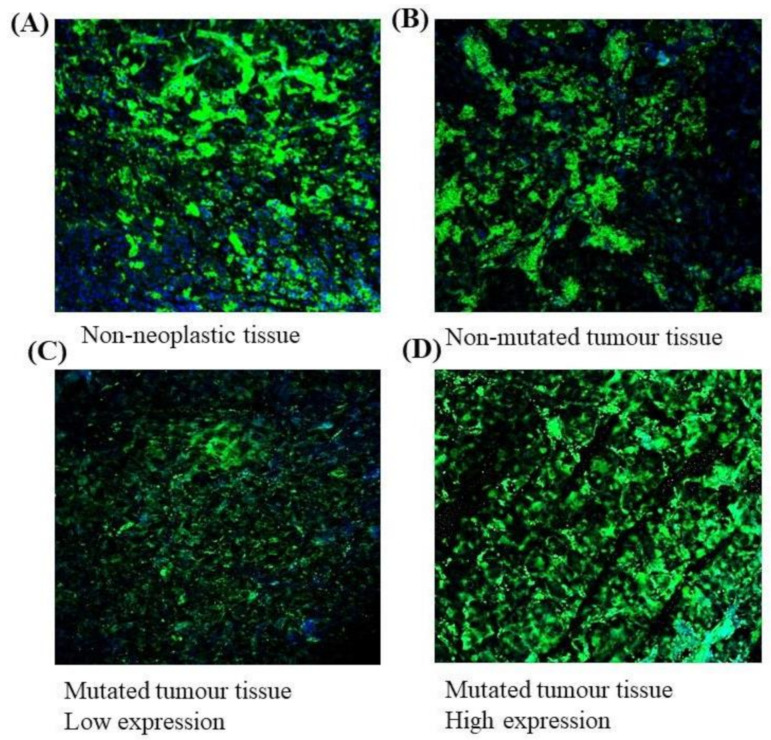
*EPAS1* protein expression in tumours and non-neoplastic tissues. Representative *EPAS1* immunofluorescence staining under confocal microscopy. (**A**) Non-neoplastic adrenal tissue. (**B**) Tumour tissue with no *EPAS1* mutation. (**C**) Mutated tumour tissue with low EPAS1 expression. (**D**) Mutated tumour tissue with high EPAS1 expression.

**Figure 5 genes-11-01254-f005:**
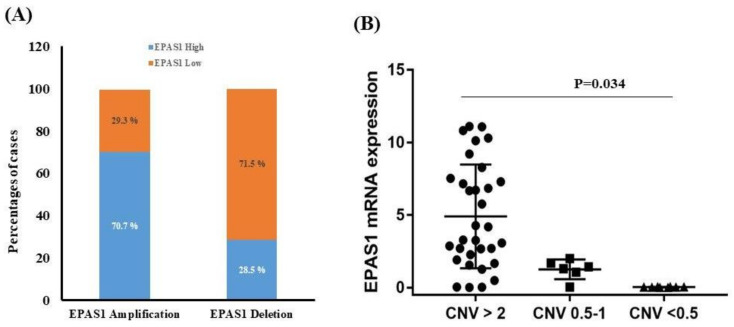
Relationship of *EPAS1* DNA number alteration and mRNA expression. (**A**) Association of *EPAS1* DNA number changes and mRNA expression. Quantitative reverse transcription polymerase chain reaction (RT-qPCR) analysis revealed that *EPAS1* DNA number amplification significantly correlated with mRNA overexpression (*p* < 0.009). (**B**) The distribution of *EPAS1* mRNA expression in patients with phaeochromocytomas/paragangliomas with a copy number of 2 or less than 2 and greater than 2. Patients with a copy number greater than 2 had shown higher mRNA expression (*p* = 0.034).

**Figure 6 genes-11-01254-f006:**
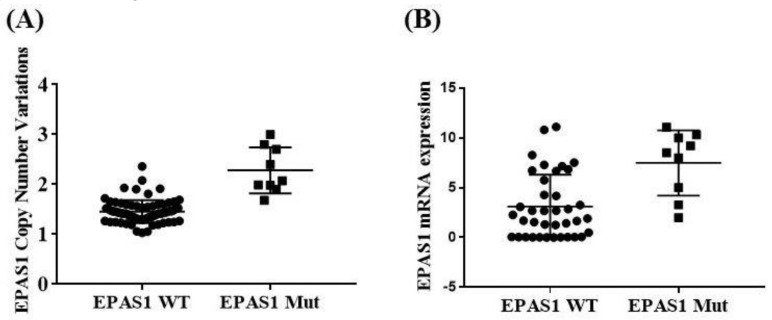
Association of *EPAS1* DNA number alteration and mRNA expression with mutations. (**A**) *EPAS1*-mutated samples had shown significant amplification of copy number in comparison to that of non-mutated samples (*p* < 0.05). (**B**) Similarly, *EPAS1*-mutated samples exhibited significantly higher expression (mRNA) when compared to that of non-mutated tissue samples (*p* < 0.05).

**Table 1 genes-11-01254-t001:** Mutations detected in the sequence of EPAS1 in phaeochromocytoma (PCC)/paraganglioma (PGL).

Sample ID	Type	Change in DNA Sequence	Change in Protein Sequence	DNA Copy Number Change	mRNA Expression	Protein Expression	Effects on Protein Features	In Silico Prediction		
								Mutation Taster	*PROVEAN*	*SIFT*
P3	PCC	c.1091A>T	p.Lys364Met	No change	High	High	Amino acids sequence change Protein structure (might be) affected Splice site changes	Diseases causing	Deleterious	Damaging
P33	PCC	c.1091A>T	p.Lys364Met	Amplification	High	High	Amino acids sequence change Protein structure (might be) affected Splice site changes	Diseases causing	Deleterious	Damaging
P78	PCC	c.1129A>T	P.Ser377Cys	Amplification	High	High	Amino acids sequence changed	Polymorphism	Neutral	Tolerated
P81	PCC	c.1091A>T	p.Lys364Met	Amplification	No Change	No Change	Amino acids sequence change Protein structure (might be) affected Splice site changes	Diseases causing	Deleterious	Damaging
P93	PCC	c.1129A>T	P.Ser377Cys	Amplification	High	High	Amino acids sequence changed	Polymorphism	Neutral	Tolerated
P94 *	PCC	c.1091A>T	p.Lys364Met	Amplification	No Change	Low	Amino acids sequence change Protein structure (might be) affected Splice site changes	Diseases causing	Deleterious	Damaging
P99	PCC	c.1091A>T	p.Lys364Met	Amplification	High	High	Amino acids sequence change Protein structure (might be) affected Splice site changes	Diseases causing	Deleterious	Damaging
P122 *	PGL	c.1091A>T	p.Lys364Met	Amplification	High	Low	Amino acids sequence change Protein structure (might be) affected Splice site changes	Diseases causing	Deleterious	Damaging

* Neurofibromatosis 1 positive.

**Table 2 genes-11-01254-t002:** Correlation of *EPAS1* mutations with clinicopathological features of patients with phaeochromocytoma and paraganglioma.

Features	Number	Mutation Positive	Mutation Negative	*p*-Value
**Total patients**	71 (100.00%)	8 (11.26%)	63 (88.74%)	-
**Gender**				
Male	37 (52.11%)	4 (10.81%)	33 (89.19%)	0.596
Female	34 (47.89%)	4 (11.76%)	30 (88.24%)	
**Age**				
≤50	40 (56.34%)	6 (15.00%)	34 (85.00%)	0.229
>50	31 (43.66%)	2 (6.45%)	29 (93.55%)	
**Race**				
Chinese	56 (78.87%)	8 (14.29%)	48 (85.71%)	0.134
Non-Chinese	15 (21.13%)	-	15 (100.00%)	
**Side**				
Unilateral	63 (88.73%)	7 (11.11%)	56 (88.89%)	0.636
Bilateral	8 (11.27%)	1 (12.50%)	7 (87.50%)	
**Tumour location**				
Adrenal gland	57 (80.28%)	6 (10.53%)	51 (89.47%)	0.497
Carotid body	14 (19.72%)	2 (12.29%)	12 (85.71%)	
**Tumour size**				
<50 mm	33 (46.48%)	1 (3.030%)	32 (96.97%)	**0.044**
≥50 mm	38 (53.52%)	7 (18.42%)	31 (81.58%)	
**Tumour weight ***				
≤50 gm	17 (45.95%)	-	17 (100.00%)	**0.000**
>50 gm	20 (54.05%)	3 (15.00%)	17 (85.00%)	
**Tumour types**				
Non-metastasizing	59 (83.10%)	5 (8.47%)	54 (91.53%)	0.167
Metastasizing	12 (16.90%)	3 (25.00%)	9 (75.00%)	

* 37 cases have tumour weight information in the present study.

**Table 3 genes-11-01254-t003:** Correlation of *EPAS1* DNA number changes with clinicopathological features of patients with pheochromocytoma and paraganglioma.

Features	Number	DNA Amplification	DNA Deletion	*p*-Value
**Total patients**	71 (100.00%)	8 (11.26%)	63 (88.74%)	-
**Gender**				
Male	37 (52.11%)	28 (75.68%)	9 (24.32%)	0.347
Female	34 (47.89%)	28 (82.35%)	6 (17.65%)	
**Age**				
≤50	40 (56.34%)	30 (75.00%)	10 (25.00%)	0.271
>50	31 (43.66%)	26 (83.87%)	5 (16.13%)	
**Race**				
Chinese	56 (78.87%)	45 (80.36%)	11 (19.64%)	0.392
Non-Chinese	15 (21.13%)	11 (73.33%)	4 (26.67%)	
**Side**				
Unilateral	63 (88.73%)	51 (80.95%)	12 (19.05%)	0.219
Bilateral	8 (11.27%)	5 (62.50%)	3 (37.50%)	
**Tumour location**				
Adrenal gland	57 (80.28%)	48 (84.21%)	9 (15.79%)	**0.037**
Carotid body	14 (19.72%)	8 (57.14%)	6 (15.79%)	
**Tumour size**				
<50 mm	33 (46.48%)	23 (69.70%)	10 (30.30%)	0.120
≥50 mm	38 (53.52%)	32 (84.21%)	6 (15.79%)	
**Tumour weight ***				
≤50 gm	17 (45.95%)	14 (82.35%)	3 (17.65%)	0.857
>50 gm	20 (54.05%)	17 (85.00%)	17 (15.00%)	
**Tumour types**				
Non-metastasizing	59 (83.10%)	47 (79.66%)	12 (20.34%)	0.852
Metastasizing	12 (16.90%)	9 (75.00%)	3 (25.00%)	

* 37 cases have tumour weight information in the present study.

**Table 4 genes-11-01254-t004:** Association of *EPAS1* mRNA changes with clinicopathological features of patients with pheochromocytoma and paraganglioma.

Features	Number	High Expression	Low Expression	*p*-Value
**Total patients**	45 (100.00%)	24 (53.33%)	21 (46.67%)	-
**Gender**				
Male	25 (55.56%)	14 (56.00%)	11 (44.00%)	0.460
Female	20 (44.44%)	10 (50.00%)	6 (50.00%)	
**Age**				
≤50	28 (62.22%)	16 (57.14%)	12 (42.86%)	0.363
>50	17 (37.78%)	8 (47.06%)	9 (52.94%)	
**Race**				
Chinese	35 (77.78%)	18 (51.43%)	17 (48.57%)	0.454
Non-Chinese	10 (22.22%)	6 (60.00%)	4 (40.00%)	
**Side**				
Unilateral	41 (91.11%)	23 (56.10%)	18 (43.90%)	0.254
Bilateral	4 (8.89%)	1 (25.00%)	3 (75.00%)	
**Tumour location**				
Adrenal gland	37 (82.22%)	24 (64.86%)	13 (35.14%)	**0.001**
Carotid body	8 (17.78%)	-	8 (100%)	
**Tumour size**				
<50 mm	17 (37.78%)	7 (41.2%)	10 (58.8%)	0.167
≥50 mm	28 (62.22%)	17 (60.71%)	11 (39.29%)	
**Tumour weight ***				
≤50 gm	6 (35.29%)	3 (50.0%)	3 (50.0%)	0.627
>50 gm	11 (64.71%)	5 (45.45%)	6 (54.55%)	
**Tumour types**				
Non-metastasizing	36 (80.0%)	22 (61.11%)	14 (38.89%)	0.057
Metastasizing	9 (20.0%)	2 (22.22%)	7 (77.78%)	

* 17 cases have tumour weight information in the present study.

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
