# Peer review of "Identification of Novel Mutations and Expressions of EPAS1 in Phaeochromocytomas and Paragangliomas"

_genes, 2020, doi:10.3390/genes11111254_

Round 1

Reviewer 1 Report

Dear authors,

This article is very interesting because authors investigated mutations and functions of EPAS1 in PPGLs.

Please confirm my comments below.

1. Authors investigated the relationship between clinicopathological findings and not only the mutations but also DNA/mRNA expression of EPAS1.

It is better to change the title of this article.

i.e. Identification of mutations and expressions of EPAS1 in pheochromocytomas and paragangliomas  

2. Authors should modify the description in lines 52-54. Dahia (2014) firstly described “two major clusters”. Authors should correct the reference of the description about “two major clusters” (change the reference or add the reference).

([Pillai et al 2016] to [Dahia 2014] or [Dahia 2014, Pillai et al 2016]>.

3. Regarding the relationship between EPAS1 mutations and clinicopathological findings, I want to ask authors the reason authors divided the patients to ≤4cm and >4cm (tumour size). Patients with ≤4cm tumour were 29 patients and those with >4cm tumour were 42 patients. There was quite a difference about the number of patients between the two groups, which might cause the controversy between the result of tumour size and that of tumour weight. I think it could be better for authors to reconsider the divided line (i.e. 5cm). If authors reconsider and the difference about the number of patients between the two groups is smaller, more proper analysis could be performed.

4. I think this study could reveal both the effect of EPAS1 mutation on clinicopathological findings and EPAS1 DNA/mRNA expressions to them.

Moreover, there was a difference between them (tumour weight and location). Authors should mention this more in discussion and conclusion. I think mutations and DNA/mRNA expressions of EPAS1 could affect PPGLs in a different manner. If it is correct, please mention. I think the difference of EPAS1 expression between phaeochromocytoma and paraganglioma (location) must be fascinating and novel information (concerning comment 1).

5. In the literature, EPAS1 mutations were also reported in Exon12. Authors should mention whether there were no mutations in Exon12 or authors did not investigate the mutations in Exon12.

6. I want to ask authors investigated the other well-known mutations such as SDHB, and so on in these samples. If authors performed, please mention.

7. Authors should correct the descriptions.

Table 1 “DAN copy number change” - “DNA copy number change”

Figure 3 “Cancer” - “phaeochromocytomas / paragangliomas”

line 346 “VEFG”- “VEGF”

In addition, I think authors wrote with British English. If it is correct, authors should modify “pheochromocytoma” to “phaeochromocytoma”.

Author Response

Response to the comments Reviewer #1

This article is very interesting because authors investigated mutations and functions of EPAS1 in PPGLs.

Response: Thank you very much for your positive comments. We really appreciate your time and efforts in reviewing our works.

 Please confirm my comments below.

  1. Authors investigated the relationship between clinicopathological findings and not only the mutations but also DNA/mRNA expression of EPAS1.

It is better to change the title of this article.

i.e. Identification of mutations and expressions of EPAS1 in pheochromocytomas and paragangliomas  

Response: Thanks for this insightful comment. The title of the manuscript modified as suggested.

  1. Authors should modify the description in lines 52-54. Dahia (2014) firstly described “two major clusters”. Authors should correct the reference of the description about “two major clusters” (change the reference or add the reference).

([Pillai et al 2016] to [Dahia 2014] or [Dahia 2014, Pillai et al 2016]>.

Response: Thanks. The references are corrected as [Dahia 2014, Pillai et al 2016]. Please see lines 52-53.

  1. Regarding the relationship between EPAS1 mutations and clinicopathological findings, I want to ask authors the reason authors divided the patients to ≤4cm and >4cm (tumour size). Patients with ≤4cm tumour were 29 patients and those with >4cm tumour were 42 patients. There was quite a difference about the number of patients between the two groups, which might cause the controversy between the result of tumour size and that of tumour weight. I think it could be better for authors to reconsider the divided line (i.e. 5cm). If authors reconsider and the difference about the number of patients between the two groups is smaller, more proper analysis could be performed.

Response: Thanks for this comment. We have reanalysed the clinical correlations using 5cm of tumour size as the dividing line as suggested. The tables and text of manuscript are updated accordingly.

  1. I think this study could reveal both the effect of EPAS1 mutation on clinicopathological findings and EPAS1 DNA/mRNA expressions to them.

Moreover, there was a difference between them (tumour weight and location). Authors should mention this more in discussion and conclusion. I think mutations and DNA/mRNA expressions of EPAS1 could affect PPGLs in a different manner. If it is correct, please mention. I think the difference of EPAS1 expression between phaeochromocytoma and paraganglioma (location) must be fascinating and novel information (concerning comment 1).

Response: Thanks. We have added more discussion; also incorporate the EPAS1 DNA/mRNA data in conclusion as suggested. Please see 481-491 and 512-513 lines.

  1. In the literature, EPAS1 mutations were also reported in Exon12. Authors should mention whether there were no mutations in Exon12 or authors did not investigate the mutations in Exon12.

Response: Thank you. In the current study, we did not screen exon 12. We will analyse exon 12 in subsequent studies. This information is given in lines 134-135.

  1. I want to ask authors investigated the other well-known mutations such as SDHB, and so on in these samples. If authors performed, please mention.

Response: We screen mutation of other known susceptible genes including SDHB using NGS and did not find any mutations in the current cohort. This information was given in discussion, please see lines 440-443.  

  1. Authors should correct the descriptions.

Table 1 “DAN copy number change” - “DNA copy number change”

Figure 3 “Cancer” - “phaeochromocytomas / paragangliomas”

line 346 “VEFG”- “VEGF”

In addition, I think authors wrote with British English. If it is correct, authors should modify “pheochromocytoma” to “phaeochromocytoma”.

Response: Thanks. All the typos are corrected accordingly.

Reviewer 2 Report

In this manuscript, Islam and colleagues describe EPAS1 mutations and alterations in DNA copy number, mRNA and protein expression in patients with PC/PGs. The authors also attempted to explore the association of abnormal EPAS1 regulation with clinical and pathological factors in PC/PGs. The authors specifically assessed EPAS1 DNA changes and mRNA expression; the authors also provide immunofluorescence assay to demonstrate EPAS1 protein expression. The authors report that in PCs, 12% of patients had 26 mutations in EPAS1 sequence; they report two novel mutations (c.1091A>T; p.Lys364Met and c.1129A>T; p.Ser377Cys) that they detected. In PGs, 7% of patients had 28 EPAS1 mutations and only one of the above mutations (c.1091A>T; p.Lys364Met) was detected. The authors state that in silico analysis shows that p.Lys364Met mutation has a pathological potential based on protein functionality, whereas p.Ser377Cys mutation is likely to be neutral or tolerated. The authors also observed that the majority of the patients had EPAS1 DNA amplification (79%; n=56/71) and 53% (n=24/45) patients shown mRNA overexpression. The authors also report a correlation between EPAS1 mutations and DNA changes, mRNA and protein overexpression, and association with tumor weight and location. This is an interesting work that follows upon a previous publication by Welander et al. 2014 and identifies a high number of genetic alterations in EPAS1 locus in PC/PG tumors. The statistically significant correlation between tumor weight and mutations is also an informative finding. Major concerns: The protein expression data in Fig.4 is qualitative and unconvincing. The authors should present a quantitative experiment to demonstrate the expression level of the protein. Minor concern: It will be interesting to know whether any other comorbidities were observed in patients.

Author Response

Response to the comments Reviewer #2

In this manuscript, Islam and colleagues describe EPAS1 mutations and alterations in DNA copy number, mRNA and protein expression in patients with PC/PGs. The authors also attempted to explore the association of abnormal EPAS1 regulation with clinical and pathological factors in PC/PGs. The authors specifically assessed EPAS1 DNA changes and mRNA expression; the authors also provide immunofluorescence assay to demonstrate EPAS1 protein expression. The authors report that in PCs, 12% of patients had 26 mutations in EPAS1 sequence; they report two novel mutations (c.1091A>T; p.Lys364Met and c.1129A>T; p.Ser377Cys) that they detected. In PGs, 7% of patients had 28 EPAS1 mutations and only one of the above mutations (c.1091A>T; p.Lys364Met) was detected. The authors state that in silico analysis shows that p.Lys364Met mutation has a pathological potential based on protein functionality, whereas p.Ser377Cys mutation is likely to be neutral or tolerated. The authors also observed that the majority of the patients had EPAS1 DNA amplification (79%; n=56/71) and 53% (n=24/45) patients shown mRNA overexpression. The authors also report a correlation between EPAS1 mutations and DNA changes, mRNA and protein overexpression, and association with tumor weight and location. This is an interesting work that follows upon a previous publication by Welander et al. 2014 and identifies a high number of genetic alterations in EPAS1 locus in PC/PG tumors. The statistically significant correlation between tumor weight and mutations is also an informative finding.

Response: We appreciate your positive feedback and thanks for your time and efforts in reviewing our current manuscript.

Major concerns: The protein expression data in Fig.4 is qualitative and unconvincing. The authors should present a quantitative experiment to demonstrate the expression level of the protein.

Response: Thank you very much for this comment. We have examined the expression of protein in 8 mutated and 8 non-mutated representative samples. The signal or intensity of protein staining generated by the protein in tissue sections under confocal microscopy were quantitated and graded according to the strength as follows. The generated signals from EPAS1 staining was categorized as "0" (0% to less than 10%), "+” (10% to <30%), “++” (30% to < 50%) and “+++” (>50%) according to the percentage and intensity of EPAS1 protein staining. Tissues in the categories of “0” & “+” were classified as No change or low, whereas “++” & “+++” were considered as high EPAS1 expression.Please see the information in lines 190-193.

Additionally, the focus of the current study is to examine the genetic alterations of EPAS1 in PPGL and their clinical significance, however, in our subsequent study we will do details quantitative analysis of EPAS1 protein expression.

Minor concern: It will be interesting to know whether any other comorbidities were observed in patients. 

Response: Thanks, one of the patient had breast cancer and another patient have myocardial infarction. This information is added in the revised manuscript (please see lines 113-114).

Reviewer 3 Report

Islam and colleagues in this paper described new EPAS1 novel mutations, alterations in DNA copy number, mRNA and protein expression in patients affected by pheochromocytomas/paragangliomas (PCCs/PGLs). The results reported are interesting and original, nevertheless, there are some points that should be addressed.

  1. In the introduction, the Authors reported that PCCs/PGLs are divided into two clusters. Nowadays it is well accepted the presence of a third cluster, that should be mention in the same section.
  2. Paragangliomas arising in the abdomen, thorax, and pelvic region usually secrete catecholamines. On the contrary, paragangliomas located in the head and neck region normally do not. In the 2.2 Clinical data of the selected cases (lines 103-104) is stated that “ All the 14 patients with PGL…..do not have catecholamines detected clinically”, but only in the Table 2 is reported that all the PGLs described in the paper are located in the carotid body. To avoid generating confusion in the reader, the Authors should specify the localization of the PGLs in section 2.2.
  3. Again, it is mentioned that catecholamines were not detected clinically in 8 patients with PCC. How were catecholamines in patients mutated for EPAS1? This point should be discussed.
  4. In figure 2, the Authors showed a representative chromatograph for the EPAS1 missense mutation, which, in that case is in heterozygosity. Was heterozygosity found in all cases or there were some patients showing a loss of it? Has been investigated if the mutations were only somatic or also germline? These two points should be discussed.
  5. In line 218, the Authors wrote that two patients EPAS1 mutated were also affected by neurofibromatosis 1. Moreover, the Authors did not find association between EPAS1 mutations with patients’ age, sex…..(lines 222-223). In lines 316-319, the Authors reported that EPAS1 mutated patients were not mutated for other susceptibility genes. However, the genetic of the patients is not mentioned. There were patients mutated for one of the other susceptibility genes? In the case, for which gene? How was EAPS1 DNA copy number, mRNA and protein expression in those patients? A partition by genes should be done.
  6. Have the Authors an Hypothesis why EPAS1 mutated tumours show high EPAS1 protein expression levels?
  7. In Table 4, the Authors showed that low expression of EPAS1 mRNA was prevalent in metastasizing tumours. This point should be extensively discussed.

Minor point

  1. Line 221. EPASI should be read as EPAS1.
  2. Lines 237-239. “Similarly, PCC showed significant higher proportion of ……..(15.97% versus 42.86%). 15.97% is lower that 42.86%.

Author Response

Response to the comments Reviewer #3

Islam and colleagues in this paper described new EPAS1 novel mutations, alterations in DNA copy number, mRNA and protein expression in patients affected by pheochromocytomas/paragangliomas (PCCs/PGLs). The results reported are interesting and original, nevertheless, there are some points that should be addressed.

 Response: Thank you very much. We appreciate your positive comment on our present work.

  1. In the introduction, the Authors reported that PCCs/PGLs are divided into two clusters. Nowadays it is well accepted the presence of a third cluster, that should be mention in the same section.

Response: Thanks for this comment. We have modified the section and added the new cluster as suggested. Please see lines 57-60.

  1. Paragangliomas arising in the abdomen, thorax, and pelvic region usually secrete catecholamines. On the contrary, paragangliomas located in the head and neck region normally do not. In the 2.2 Clinical data of the selected cases (lines 103-104) is stated that “ All the 14 patients with PGL…..do not have catecholamines detected clinically”, but only in the Table 2 is reported that all the PGLs described in the paper are located in the carotid body. To avoid generating confusion in the reader, the Authors should specify the localization of the PGLs in section 2.2.

Response: Thanks. We have added the information as suggested. Please see lines 106-107.

  1. Again, it is mentioned that catecholamines were not detected clinically in 8 patients with PCC. How were catecholamines in patients mutated for EPAS1? This point should be discussed.

Response: Thank you very much for this comment. Six out of eight patients with EPAS1 mutation had shown catecholamine secretion. This information is added in the revised manuscript. Please see lines 108-109.

  1. In figure 2, the Authors showed a representative chromatograph for the EPAS1 missense mutation, which, in that case is in heterozygosity. Was heterozygosity found in all cases or there were some patients showing a loss of it? Has been investigated if the mutations were only somatic or also germline? These two points should be discussed.

Response: Thank you very much for this insightful comment. Yes, you are correct. All the eight cases show heterozygosity and no homozygous loss was detected. We have examined only somatic mutation no germline mutations were investigated. This information is added in the results and discussions sections. Please see lines 204 and 436.

  1. In line 218, the Authors wrote that two patients EPAS1 mutated were also affected by neurofibromatosis 1. Moreover, the Authors did not find association between EPAS1 mutations with patients’ age, sex (lines 222-223). In lines 316-319, the Authors reported that EPAS1 mutated patients were not mutated for other susceptibility genes. However, the genetic of the patients is not mentioned. There were patients mutated for one of the other susceptibility genes? In the case, for which gene? How was EAPS1 DNA copy number, mRNA and protein expression in those patients? A partition by genes should be done.

Response: Thanks for this comment. We have screened other susceptible genes in patients having EPAS1 mutations using NGS and noted that only the two patients with Neurofibromatosis 1 had shown NF1 mutations. No other genes have any mutations. This information is given in lines 440-443.

The EPAS1 copy number, mRNA and protein expression of the two patients with NF1 mutations and EPAS1 mutations are shown in table 1. There is no consistent relationship among these parameters.

  1. Have the Authors an Hypothesis why EPAS1 mutated tumours show high EPAS1 protein expression levels?

Response: Thanks. As most of the tumour samples (78.87%; table 3) exhibited EPAS1 DNA amplification, thus, the mutated samples could show protein overexpression (5 out of 8) as a result of abundance DNA copy and higher mRNA level. As we have noted most of mutated tumour exhibited DNA amplification and high mRNA expression.

Thus, we hypothesize that the mutations could induce a gain-of-function of the protein by increasing their stability of the protein and/or decreasing the ubiquitin mediated degradation. In the tumour microenvironment, the mutations may be associated with the reduced oxygen desensitizer for the proteins, which in turn slows down the breakdown process of EPAS1 protein. Since, EPAS1 act as an oxygen sensitive components of HIFs, thus, these mutation may be associated with reduced oxygen response of the protein, thereby slowing their breakdown, resulting in higher protein levels in mutated samples. Previous study has reported this, please see lines 503-506.       

  1. In Table 4, the Authors showed that low expression of EPAS1 mRNA was prevalent in metastasizing tumours. This point should be extensively discussed.

Response: Thanks a lot. We have added more discussion regarding EPAS1 mRNA and tumour metastasis. Please see lines 491-495.

Minor point

  1. Line 221. EPASI should be read as EPAS1.
  2. Lines 237-239. “Similarly, PCC showed significant higher proportion of ……..(15.97% versus 42.86%). 15.97% is lower that 42.86%.

Response: Thanks, these typos were corrected accordingly.

Round 2

Reviewer 2 Report

This should be adequate.

Reviewer 3 Report

The Authors modified the paper, and made all the changes requested.